# The Environmental Consequence of Early Colorectal Cancer Detection: A Literature Review of the Environmental Impact Assessment of Colorectal Cancer Diagnostic Pathways

**DOI:** 10.3390/ijerph22111649

**Published:** 2025-10-30

**Authors:** Ifeoluwa Osinkolu, Arron Lacey, Dean Harris

**Affiliations:** Faculty of Medicine, Health and Life Sciences, Swansea University, Swansea SA2 8PP, Wales, UK; a.s.lacey@swansea.ac.uk (A.L.); d.a.harris@swansea.ac.uk (D.H.)

**Keywords:** colorectal, cancer, life cycle assessment, sustainability

## Abstract

Background: Colorectal cancer (CRC) is a burden to healthcare globally, with early detection vital to improving outcomes. While screening and early diagnostic strategies are being widely implemented, their environmental impact remains underexplored. The purpose of this literature review is to examine the existing research on the environmental footprint of non-emergency sporadic CRC diagnostic pathways and provide an overview of environmental impact assessment processes. Principal findings: Population-based screening appears environmentally beneficial over time, but its efficiency critically determines its net impact. Studies identify endoscopy as having the highest environmental impact among testing modalities. The dominant contributor to this is patient and staff travel. By contrast, faecal-based tests appear to have the lowest environmental footprint. Notably, pathway-wide assessments are limited, and methodological inconsistencies hinder comparing studies. Conclusions: There is an urgent need to standardise a healthcare sector-specific framework for environmental impact assessments. Emerging biomarker-based diagnostics will require a robust pathway-wide environmental impact assessment before clinical integration.

## 1. Introduction

Colorectal cancer is the 3rd most common cancer in the world, with its incidence expected to rise [1]. It is the 4th most common cancer in the UK, with 44,000 new cases diagnosed every year [2]. The pathogenesis of sporadic colorectal cancer is theorised to follow the adenoma carcinoma pathway. This is a process which is characterised by the development of a precursor lesion (such as an adenomatous polyp or sessile serrated lesion) prior to malignant transformation [3]. This presents an opportunity for prevention by removing precursor lesions and early-stage identification. As it stands, endoscopy (flexible sigmoidoscopy or colonoscopy), which involves direct visualisation, biopsy, and the potential excision of a lesion are the main investigative techniques utilised for diagnosing colorectal cancer. Alternatively, computer tomography colonography (CTC) serves as an imaging modality for patients who are not able undergo endoscopy. Positive findings on CTC may subsequently lead to colonoscopy for definitive diagnosis.

Colorectal cancer may present through a symptomatic manifestation or asymptomatic screening. Symptoms of colorectal cancer are diverse, with common presentations including iron-deficiency anaemia, unintentional weight loss, abdominal pain and blood in stools [4]. To aid the challenge of early diagnosis in symptomatic patients, clinical pathways are being established to fast-track these patients for investigation with speciality services. Despite the use of symptom triage tests such as the faecal immunochemical test (FIT), the number of resultant normal colonoscopies remains unacceptably high. Colorectal cancer screening refers to the use of diagnostic tests and procedures in asymptomatic individuals with the aim of detecting precancerous lesions or early-stage colorectal cancer [5]. Population-based screening is recommended in many countries with variable uptake [5].

The link between healthcare delivery and its negative environmental impact is well documented [6]. This recognition has prompted healthcare systems worldwide to place a greater emphasis on environmental sustainability [7]. Although substantial research efforts have been directed towards developing new highly efficient colorectal cancer diagnostic modalities with superior efficacy, such as liquid biopsies and other biosensor technologies, the environmental implications of integrating these testing tools into existing clinical pathways remain largely underexplored. Addressing this requires a thorough evaluation of the environmental impact associated with the current colorectal cancer diagnostic pathways.

## 2. Aim and Objectives

This review aims to assess the environmental footprint of non-emergency sporadic colorectal cancer diagnosis by synthesising the existing literature on the subject. The objectives are as follows:Outline the methodological foundations of environmental impact assessments.Identify factors that impact the environmental footprint of current colorectal cancer diagnosis.Highlight areas for future research.

## 3. Methodology

No formal literature review framework (such as PRISMA) was used for this review. An extensive search was conducted across major academic databases using a combination of key terms like “carbon footprint colorectal cancer,” “environmental impact of colorectal cancer,” and “life cycle assessment of colorectal cancer screening”. The number of directly relevant studies was limited (*n* = 3). As a result, papers addressing related concepts have been incorporated to enrich the review. Selected articles were appraised based on their relevance, methodological clarity, and their contribution to the topic, rather than formal quality scoring. Themes and patterns noted in the literature were analysed in relation to the objectives of the review. This approach allowed for a comprehensive exploration of the topic despite the limited availability of focused studies.

## 4. Approaches to Environmental Impact Analysis

The most comprehensive process to evaluate the environmental impact of a product or service is a life cycle assessment (LCA) [8,9,10]. Integrated into the LCA process is the accounting of potential environmental consequences across all life cycle stages of a product, from the extraction of raw material, production, and use, to the disposal of the product (cradle-to-grave analysis). The principles, framework, requirements, and guidelines for performing an LCA have been standardised by the International Organisations of standardisation ISO 14040:2006 [11] and ISO 14044:2006 [12]. An LCA process is typically structured into four distinct phases as shown in Figure 1: goal definition and scoping, inventory analysis, impact assessment, and interpretation [11].

In the goal definition and scoping phase, the purpose of performing the LCA is clearly stated; the intended audience and scope of study are also set out. In the inventory analysis phase, quantitative data is collected on all inputs and outputs associated with each stage of the product or service life cycle. Primary inventory data (directly measured) may be used, or secondary data (already existing data from previous researchers) from databases may be used. The impact assessment phase is where the choices of the environmental impact factors to be measured are made, and where inventory data is converted to environmental impact factors. After this comes the interpretation phase, where environmental impact hotspots are identified, a sensitivity and uncertainty analysis is carried out, and conclusions are drawn [11].

The process used to translate input flows, resources, and emissions data gathered from inventory analysis into quantifiable environmental impact scores is known as the life cycle impact assessment method (LCIA). Within an LCIA, characterisation factors are the environmental impact per unit of a given stressor. Characterisation factors can be derived at two levels: midpoint and endpoint. Midpoint characterisation focuses on specific points along the cause-to-impact pathway and is more directly linked to environmental flows, leading to a lower uncertainty. In contrast, endpoint characterisation assesses the broader impacts on human health, ecosystem quality, and resource scarcity. Endpoint characterisations are easier to interpret, but their results are more uncertain [13]. One of the most widely used LCIA methods is the ReCiPe 2016. Originally developed in 2008 through a collaboration between the Dutch National Institute for Public Health and the Environment, Radbound University Nijmegen, Leiden University, and PRe consultants, it was updated in 2016 to improve the applicability of its characterisation factors, address previous limitations, and add new damage pathways [14]. ReCiPe 2016 translates inventory analysis data into 18 midpoint impact categories and 3 endpoint impact categories as shown in Figure 2.

Other LCIA methods are available. The Environmental footprint 3.1 method (EF 3.1) [15], developed by the European commission under its product and organisational environmental footprint initiatives, is widely used for European Union-based assessments and regulatory compliance. It covers midpoint categories such as climate change, toxicity, acidification, and resource use. The Intergovernmental panel on climate change [16] method provides standardised assessments of climate change impact using global warming potential. This method enables organisations to understand the impact of their products on the climate over different time periods. The Centrum voor Milieukunde Leiden (CML) method [17] developed by Leiden University in the Netherlands focuses on midpoint indicators like global warming potential and acidification potential. It is commonly used in academic research and industrial applications. In North America, the Tool for the reduction and assessment of chemical and other environmental impacts (TRACI) [18] method, developed by the U.S. Environment Protection Agency, is regularly used for regulatory compliance and sustainability reporting of midpoint impact categories. The choice of LCIA methodology is guided by the impact categories to be assessed (midpoint or endpoint), as defined in the goal and scope of the project. In addition, sector-specific guidance and the latest scientific evidence or consensus are important considerations in determining the approach.

Conducting a life cycle analysis is complex, resource-intense, and time-consuming, which may sometimes render a comprehensive analysis impractical. The approach to life cycle assessment was originally developed for industrial application. This has allowed for easy adaptation when used for quantifying the environmental impact of individual healthcare products, pharmaceuticals, and devices. However, as efforts grow to further understand the broader environmental impact of healthcare systems, clinical pathways, and processes, conventional LCA methodologies have undergone significant adaptations. More often in healthcare, environmental impact assessments often employ simplified approaches to increase feasibility. A common adaptation involves limiting the environmental impact categories analysed. Carbon footprinting studies (isolated greenhouse gas emissions), waste studies, and water usage evaluations are examples of this. The carbon footprinting approach is particularly prevalent in healthcare research. Another widely common adaptation is performing an economic input–output LCA, which utilises economic data (monetary value of inputs and outputs) to estimate environmental impact. This approach was applied by Steenmeijer et al. (2022) to estimate the environmental impact of the Dutch healthcare sector [19]. The predominant strategy in healthcare LCA is the hybrid approach. This combines simplified LCA techniques that limit environmental impact indicators and target key life cycle stages with a hybrid inventory analysis strategy, incorporating both primary and secondary data sources and the economic input–output LCA methodology. This enables a more robust estimation of the selected environmental impact indicators. However, the absence of a standardised approach to environmental impact assessment in healthcare poses a major challenge. The wide variation in the analytical approaches and reporting has made comparing research studies and generalising study outcomes impractical [20].

A recent scoping review to identify all the environmental impact factors currently measured in healthcare has highlighted considerable inconsistencies regarding which environmental impact factors are measured, the unit of measurement, and the methodologies used for assessments [20]. However, the review was able to group the environmental impact factors into three main categories; pollutant emissions (e.g., greenhouse gases), waste generation, and resource use [20].

## 5. Early Colorectal Cancer Diagnosis and the Environment

The goal of early colorectal cancer diagnosis is to improve survival rates and reduce colorectal cancer disease burden. To achieve this, various diagnostic modalities are strategically deployed into clinical pathways, either as stand-alone tools or in combination. Endoscopy (colonoscopy and flexible sigmoidoscopy) remains the gold standard modality for colorectal cancer diagnosis, with a high sensitivity for detecting early diseases and precancerous lesions. However, it is an invasive and costly procedure, and endoscopy services are frequently under significant pressure to meet the growing demand for their services [21]. Non-invasive faecal-based tests such as the guaiac faecal occult blood test and faecal immunochemical test (FIT) are also commonly utilised. Stool DNA with faecal immunochemical tests (sDNA-FIT) is also growing in popularity [22]. While these tests are non-invasive and cheaper, they have a lower sensitivity to high-risk precancerous lesions when compared to colonoscopy. Computer tomographic colonography (CTC) is another diagnostic modality commonly utilised; however, it is limited by the need for bowel preparation, radiation exposure, and low sensitivity for small lesions [23]. Emerging diagnostic modalities focus on detecting colorectal cancer biomarkers like circulating DNA, circulating tumour cells, and microRNAs, proteins, and other molecular markers in blood samples. Although promising, these tests are not yet routinely utilised in clinical practice [24,25].

Internationally, population-based screening represents the most widely implemented strategy for the early detection of asymptomatic colorectal cancer [5]. These programs employ a range of testing strategies, including faecal-based tests (faecal occult blood test, faecal immunochemical test, FIT, and stool DNA with FIT [sDNA-FIT]), endoscopy (colonoscopy and flexible sigmoidoscopy), computed tomography colonography, or a combination of these tests [5]. Screening approaches fall into two categories: organised and opportunistic. Organised screening involves a dedicated program with a process of invitation and a follow-up protocol for the target population, while opportunistic screening is conducted on an ad hoc basis outside of formal programs. There is an international variation in both implementation strategies and testing modalities employed in population-based screening [5]. The UK’s national bowel screening program, established in 2006, currently offers biennial faecal immunochemical testing for individuals aged 51–74, with positive tests triggering a colonoscopy or CTC [26]. Meanwhile, in the United States, opportunistic screening is employed, guided by the US Preventative services Task Force recommendations for multiple screening modality options for 45–75-year-olds. These recommendations include a FIT or sDNA-FIT every year, CTC every 5 years, or colonoscopies every 10 years [27]. Despite geographical variation in implementation, all bowel cancer screening programs share the objective of improving patient outcomes through the earlier detection of cancers and precancerous lesions. The impact of screening on the incidence of colorectal cancer is dependent on the screening modality deployed as well as the population uptake [5]. Notably, countries that offer an initial combined colonoscopy and faecal testing strategy have demonstrated a decrease in their colorectal cancer incidence when compared to countries that only offer faecal-based screening [5].

An alternate strategy for the early detection of sporadic colorectal cancer involves fast-tracking investigations and the clinical assessment of symptomatic patients. In the United Kingdom, this occurs through the Urgent Suspected colorectal cancer (USC) referral pathway. This pathway allows general practitioners (GPs) to expedite secondary care investigations for patients presenting with non-specific symptoms suggestive of colorectal cancer [28]. GPs operate within the framework of established referral guidelines, which have minor variations across the four UK nations. Following clinical assessment, a full blood count, and faecal immunochemical testing (FIT), the need for a USC referral is determined. A direct referral for colonic visualisation (straight-to-test) through endoscopy or CTC may also be performed in patients with an iron deficiency anaemia and positive FIT. This pathway ultimately streamlines referrals for patients with possible colorectal cancer to ensure the timely review by a colorectal specialist, typically supplemented by a luminal visualisation through either a colonoscopy or CTC. Versions of this pathway exist in other countries [29,30].

To quantify the environmental implications of population-based colorectal cancer screening, Yousef et al. (2024) conducted a study that demonstrated that colorectal cancer screening has an overall positive environmental impact [31]. The researchers performed a longitudinal carbon footprint assessment of screening colonoscopies in the U.S, estimating the savings in carbon emissions attributable to early cancer detection and prevention. Their methodology involved calculating the anticipated number of patient visits for cancer treatment and surveillance avoided over a ten-year period as a result of one year of screening colonoscopies. These findings were translated into avoided travel distances in miles, and subsequently into carbon emission savings using the U.S. Environmental Protection Agency’s travel emission factors [31]. Notably, their study did not account for potential environmental contributors such as devices, equipment, energy consumption, staff travel, or waste generated during screening. Despite these limitations, their results highlight the substantial environmental benefit screening could have in reducing patient travel, with one year of screening preventing 395 million miles of patient travel over a decade [31].

Some studies have undertaken a comparative analysis of the various testing modalities deployed in population-based screening. Rudrapatna et al. (2025) performed a comparative environmental impact assessment of three screening modalities (FIT, colonoscopy, and CTC) in California, USA [32], where colonoscopy represents 80% of screening utilisation. Their study goal was to identify the screening modality with the best environmental profile. Employing a hybrid LCA methodology that incorporated patient and staff travel, equipment, consumables, waste, and energy used, they quantified global warming potential and damage to human health. The results estimated per-test emissions of 0.27 kgCO_2_e (FIT), 28.8 kgCO_2_e (CTC), and 43.3 kgCO_2_e (colonoscopy), with the associated disability-adjusted life year (DALY) reductions of 0.002, 0.25, and 0.35, respectively [32]. Even after accounting for a follow-up colonoscopy after every positive FIT result, their 10-year longitudinal impact analysis favoured the use of FIT over the other modalities. This study identified fuel-based travel as the largest determinant of environmental impact, as colonoscopies and CTCs require patient travel, whereas FIT relies on courier services [32]. A separate study conducted by Alcock et al. (2025) [33] compared the carbon footprint of triennial stool DNA-FIT (sDNA-FIT) testing to decennial colonoscopy in a simulated cohort of 1 million average risk individuals aged 45–75. Their methodology used published resource data, US environmental protection agency emissions calculators, patient and freight travel estimates, and the colorectal cancer and adenoma incidence and mortality (CRC-AIM) microsimulation model to calculate the emissions from screening procedures, follow-ups, and associated logistics. Their findings showed that colonoscopy-based screening generated 59% more greenhouse gas emissions than the sDNA-FIT strategy primarily due to higher energy, waste, and travel requirements [33].

Beyond the context of population-based screening, several studies have assessed the environmental impact of endoscopy. Lammer et al. (2025) conducted a single-centred observational study to evaluate the environmental impact of diagnostic colonoscopy in Radboud University medical centre in the Netherlands by performing a life cycle assessment of colonoscopies performed in their service [34]. Their robust methodology gathered data on all recourses used, mapped across all life cycle stages from raw material extraction to waste disposal. Using the Ecoinvent database and ReCiPe 2016 LCIA methodology, they showed that a single colonoscopy generates 56.4 kg of CO_2_ equivalent emissions and causes a loss of approximately 1 h of healthy life. The transportation of patients and staff contributed to 76.5% of the environmental burden, followed by the use of disposable products [34]. Lacroute et al. (2023) performed a carbon footprint analysis of gastrointestinal endoscopy in Strasburg, France [35], using a hybrid LCA methodology accounting for energy consumption, medical gases, equipment, consumables, freight, travel, and waste. They estimated the carbon emissions per endoscopic procedure to be 28.4 kgCO_2_e, with 45% attributable to patient and staff travel using internal combustion cars [35]. These findings align with Rughawani et al. (2025) in India [36], who reported 38.45 kgCO_2_e per endoscopic procedure (higher for therapeutic procedures). Again, patient travel was identified to be the source of 83% of emissions [36]. Endoscopy has also been identified as a major source of hospital waste generation [37], with waste per procedure ranging internationally from 0.5 kg (India) to 3.03 kg (U.S.), and a significant proportion ending up in landfills [36,38].

Although studies that specifically evaluate the environmental impact of CT colonography are lacking, those on general CT imaging can offer valuable insights. McAlister et al. (2022) in Australia determined that a single CT scan produces 9.2 kgCO_2_e, with 91% resulting from electricity use [39]. This LCA study excluded energy consumption by heating, ventilation and air conditioning, and scanner manufacturing, making their findings comparable internationally. CTCs likely exceed these values due to bowel preparation and the additional consumables required. Another critical environmental concern not factored in the study is water contamination from iodinated contrast media (ICM). Water waste treatment plants lack the ability to remove iodine contrast media; this has led to an increasing ICM presence in sources of drinking water and surface water [40]. The potential cytotoxic and genotoxic effects of ICM by-products raise further ecological and public health concerns [40].

A comprehensive study that accounts for all the interactions across multiple services (GP, blood test services, specialist clinical appointments, pathology, etc.) along a patient’s diagnostic journey is lacking. Additionally, an evaluation of how diagnostic modalities are integrated and utilised within clinical pathways is also lacking. So far, studies focus on comparing individual testing modalities.

## 6. Discussion

Although the body of literature dedicated to evaluating colorectal cancer diagnosis through the lens of environmental impact remains limited and methodologically inconsistent, this review provides some valuable insight. Colorectal cancer screening demonstrates a positive environmental impact over time, primarily due to the early detection and prevention of cancer, which reduces the need for more intensive healthcare intervention. Colorectal cancer testing modalities that require less patient and staff travel exhibit a lower environmental footprint. This largely accounts for the higher environmental impact of colonoscopy and CTC compared to stool-based tests. There remains a gap in the research evaluating the cumulative environmental impact across the entire diagnostic journey, from initial contact to definite diagnosis.

Environmental impact assessment studies in healthcare remain in their early stages, without a widely agreed-upon framework to standardise an industry-specific methodology. This presents challenges in evaluating the quality of existing studies and comparing findings. Compounding this issue is the low number of healthcare professionals with both a deep understanding of clinical pathways and the technical expertise required to conduct environmental impact assessments. These studies are often resource-intensive and time-consuming, which may explain the bulk of the literature focusing on single procedures or isolated patient encounters, even when simplified adaptations of traditional life cycle assessment methodologies are employed. While assessing individual points along a patient’s journey can offer valuable insights and opportunities for environmental improvements, this approach risks overlooking the broader impacts to healthcare systems. A comprehensive approach evaluating entire diagnostic pathways could offer crucial guidance for redesigning healthcare services in a more sustainable manner. This holistic perspective will be essential to achieve the meaningful decarbonisation of our healthcare provisions.

Colorectal cancer causes a significant burden to healthcare systems in the world [1], and efficient early detection is bound to help mitigate this. As demonstrated by Yousef et al. (2024) [31], a reduction in healthcare burden can be translated to environmental benefits. In principle, programs dedicated to early detection should result in net environmental gains. However, this may not always be the case. Despite the increase in the utilisation of both the Urgent Suspected Cancer (USC) pathway and bowel cancer screening programs in the UK, the current epidemiological data fail to suggest a significant impact on the incidence of colorectal cancer by stage. Notably, the expanded utilisation of the USC pathway has led to the gradual decline in its diagnostic yield, with its conversion rate (proportion of referrals leading to a cancer diagnosis) declining from 5.4% in 2011/12 to 2.8% in 2022/23 [41]. Within the screening program, 6500 cancer cases were detected from 88,392 diagnostic procedures (7.4% yield) in 2021/22 [42]. These trends, as documented by Cancer Research UK, suggest an inefficiency that may offset the environmental benefits of early detection. The design and operational efficacy of early detection programs are therefore critical. A program with low efficacy risks generating high volumes of potentially avoidable healthcare activity, leading to the needless consumption of finite resources and an increased environmental impact. This issue is particularly important in the UK, where post-pandemic pressures on endoscopy services have led to substantial delays in meeting both routine and surveillance target times nationally [21].

Patient and staff travel has consistently emerged as the single largest contributor to the environmental impact of endoscopic procedures [34,35,36], leading to endoscopy being the diagnostic modality with the highest environmental impact. Across all the studies reviewed, a common assumption is travel occurring exclusively with internal combustion engine vehicles. In reality, travel behaviours vary significantly both nationally and internationally, ranging from the mode of transport to vehicle type. The increasing use of electric vehicles per population was not modelled into the reviewed LCA assumptions. Strategies to alleviate the impact of travel need to factor in local contexts. Urban areas with good public transport infrastructure can design pathways around public transport use, whereas rural regions may need to focus more on encouraging environmentally efficient private transport options or integrating virtual consultations, where feasible, to reduce patient travel altogether. Importantly, robust life cycle assessments are able to model and compare different transportation scenarios within the same clinical pathway. An analysis of this can inform decision-makers involved in clinical pathway design, enabling the development of environmentally optimised services suited to the local population needs.

Emerging colorectal cancer diagnostic modalities have the potential to transform early colorectal cancer detection. Non-invasive, more accurate modalities capable of detecting colorectal cancer biomarkers (protein, DNA, RNA, microRNA, and other molecules) in blood and stools are the focus of much research [43]. Although these emerging tests promise a superior sensitivity and specificity compared to the currently used non-invasive testing modalities, the strategy for deployment for widespread clinical use must be guided by meticulous environmental impact assessments. For instance, a blood-based test with a superior positive predictive value than colonoscopy could significantly reduce the number of endoscopies performed, suggesting a clear environmental benefit. However, if such a test requires patient travel to secondary care facilities for blood sampling, these potential environmental gains could be significantly offset. Travel has consistently been found to be the single largest contributor to the environmental impact of colonoscopies. An LCA may reveal testing in the primary care setting to be associated with less travel and therefore result in lowering the environmental burden. Understanding the environmental impact not only of these tests themselves but also the clinical pathways in which they are to be embedded will be essential to ensure that the clinical benefits these tests offer are not undermined by unintended environmental harm.

Our current understanding of the environmental impact associated with colorectal cancer diagnosis remains limited, but the consideration of mitigation strategies can be informed by the principles of sustainable surgical practice [44]. The primary prevention of colorectal cancer through modifiable risk factor reduction, like decreased tobacco and alcohol use, reduced red meat consumption, increased physical activity, and obesity management, offers the most significant opportunity to lessen both disease burden and downstream resource utilisation [45]. Early education through public health initiatives targeting these factors could result in substantial long-term benefits. Effective screening is an important secondary prevention tool that enables the detection and removal of precancerous lesions and early-stage malignancies [46]. However, reducing inefficiencies in the screening approaches can present further opportunities for environmental optimisation. Patient travel can be reduced by leveraging technology that facilitates virtual clinics [47]. Endoscopy offers multiple avenues for environmental impact savings. Reducing biopsy jar use, appropriate waste segregation protocols, adopting energy saving measures, and transitioning to reusable devises and PPE where feasible are all strategies to reduce the environmental impact of endoscopic procedures [48].

Implementing sustainable change within healthcare systems is challenging, requiring the balancing of competing priorities. The FIT versus colonoscopy for screening debate is an example of tensions between environmental benefits (lower emissions with FIT) and clinical performance (higher sensitivity with colonoscopy). Economic considerations further complicate decision-making, with budget-constrained healthcare systems. To support informed and balanced decision-making, more studies are needed that rigorously assess the environmental impact of entire clinical pathways. As part of the sustainable transition of our healthcare systems, pathway designs would need to be guided by the triple bottom line framework [49], weighing patient benefits against environmental, social, and economic costs. This will be the driving force for achieving sustainable transformation in colorectal cancer diagnosis and in healthcare more broadly.

## 7. Conclusions

Colorectal cancer remains a global health burden with rising incidence. Strategies such as population-based screening and streamlined referral pathways for suspected cancer cases are increasingly employed to facilitate early diagnoses. They utilise testing modalities such as endoscopy, faecal immunochemical testing (FIT), and CT colonography. Although dedicated studies assessing the environmental impact of these strategies are limited, evidence suggests that population-based screening can result in long-term environmental benefits. The degree of environmental benefit is dependent on how efficient the program is in the early diagnosis and detection of precancerous lesions. Endoscopy has consistently been identified as the diagnostic modality with the highest environmental impact, largely due to patient and staff travel. Further studies that comprehensively assess the environmental impact of a patient’s entire colorectal cancer diagnostic journey are required. Moreover, environmental impact assessment studies in healthcare also require sector-specific standardisation to ensure consistency and comparability across studies. As new, more accurate, and less invasive diagnostic technologies emerge, integration into clinical use should be guided by robust environmental evaluations. This will ensure that their clinical potentials are maximised without derailing the ambition of achieving environmentally sustainable healthcare delivery.

## Figures and Tables

**Figure 1 ijerph-22-01649-f001:**
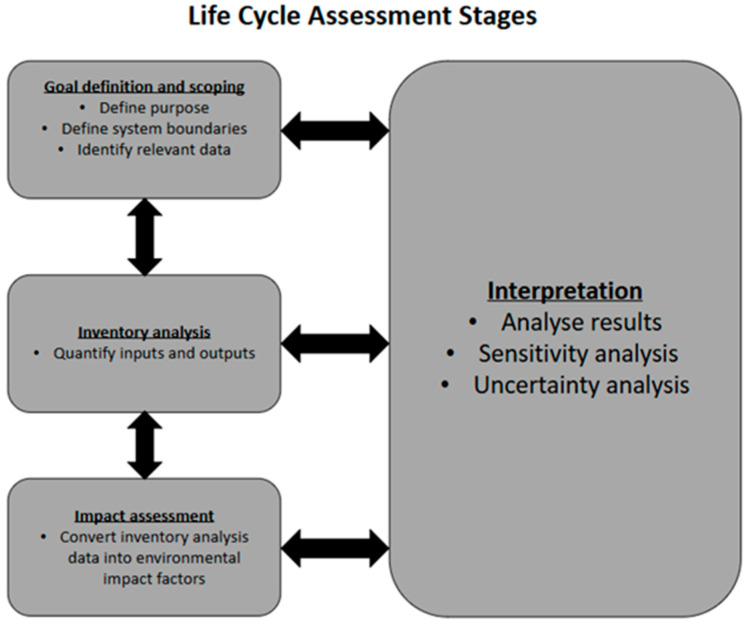
Life cycle assessment stages according to ISO standards [11,12].

**Figure 2 ijerph-22-01649-f002:**
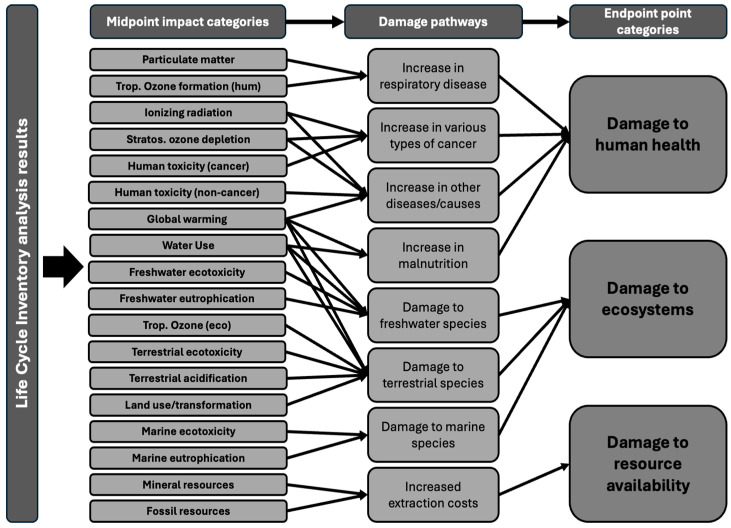
Framework of the ReCiPe 2016 showing the link between inventory analysis results, midpoint impact category, and endpoint impact categories from [14].

## Data Availability

No new data were created or analyzed in this study. Data sharing is not applicable to this article.

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
