# Peer review of "The Environmental Consequence of Early Colorectal Cancer Detection: A Literature Review of the Environmental Impact Assessment of Colorectal Cancer Diagnostic Pathways"

_ijerph, 2025, doi:10.3390/ijerph22111649_

Round 1

Reviewer 1 Report

Comments and Suggestions for Authors

ijerph-3799314

The Environmental consequence of early colorectal cancer detection: A literature review of the environmental impact assessment of colorectal cancer diagnostic pathways.

Review article

Comments to the author(s)

Dear author(s)

  1. The author is appreciated for selecting a relevant and timely topic for the review; however, the manuscript lacks comprehensive coverage in several critical areas, necessitating substantial revision.
  2. A concise and factual abstract is required, clearly stating the purpose of the review, the principal findings, and the major conclusions.
  3. In the introduction, while the author provides substantial information on the basic concepts, statistical reports, and diagnostic procedures related to colorectal cancer, the section fails to explicitly present the aim, objectives, and research questions of the review. These should be clearly articulated—preferably in bullet points—for better clarity. An overarching overview is also recommended to enhance navigability.
  4. In Section 2, beyond the life cycle assessment, the inclusion of prognosis assessment approaches for colorectal cancer and methodologies for environmental impact analysis is advised.
  5. Section 3 is limited in scope and references. The author has relied on very few resources, and the content is too brief. Numerous recent research and review articles on early colorectal cancer diagnosis have been published in MDPI and MDPI reputable sources; integrating these would strengthen the manuscript and improve its relevance.
  6. The discussion section entirely lacks reference citations. A comparative discussion is required, linking back to the research questions stated in Section 1, and emphasizing the societal impact of the findings.
  7. Throughout the manuscript, no figures or tables are included. It is recommended to incorporate relevant visual elements, particularly a PRISMA flow diagram, to enhance clarity and structure.
  8. The manuscript is notably short for a review article. Both the length and depth of coverage should be increased. Currently, the article cites only 26 references, which is insufficient for a review of this scope. The inclusion of additional, recent, and directly relevant literature is strongly recommended.

Author Response

Thank you for your comments. Please find below our responses

 Comment: A concise and factual abstract is required, clearly stating the purpose of the review, the principal findings, and the major conclusions.

Response: We agree with your comment and have revised the abstract to clearly state the purpose of the review, the principal findings, and the major conclusions.

Comment: In the introduction, while the author provides substantial information on the basic concepts, statistical reports, and diagnostic procedures related to colorectal cancer, the section fails to explicitly present the aim, objectives, and research questions of the review. These should be clearly articulated—preferably in bullet points—for better clarity. An overarching overview is also recommended to enhance navigability.

Response: We agree with your comment and have made substantial revisions to the introduction, including a clear articulation of the aims and objectives in bullet points, along with an overarching overview to improve navigability.

Comment: In Section 2, beyond the life cycle assessment, the inclusion of prognosis assessment approaches for colorectal cancer and methodologies for environmental impact analysis is advised.

Response: Thank you for your comment. Section 2 focuses on environmental impact assessments in the healthcare sector, comparing them to pan-industry international standards. Currently, there are no established methodologies that link cancer prognosis assessment directly to environmental impact analysis. We have clarified this in the revised manuscript.

Comment: Section 3 is limited in scope and references. The author has relied on very few resources, and the content is too brief. Numerous recent research and review articles on early colorectal cancer diagnosis have been published in MDPI and MDPI reputable sources; integrating these would strengthen the manuscript and improve its relevance.

Response: We agree with your comment and have significantly expanded Section 3 by incorporating additional recent research to enhance the scope and relevance of the manuscript.

Comment: The discussion section entirely lacks reference citations. A comparative discussion is required, linking back to the research questions stated in Section 1, and emphasizing the societal impact of the findings.

Response: We agree with your comment and have thoroughly revised the discussion section to include reference citations and a comparative discussion linked to the research questions.

Comment: Throughout the manuscript, no figures or tables are included. It is recommended to incorporate relevant visual elements, particularly a PRISMA flow diagram, to enhance clarity and structure.

Response: We agree with your comment. Two figures, have been added to the manuscript. A methodology section has also been included to improve clarity and structure of approach.

Comment: The manuscript is notably short for a review article. Both the length and depth of coverage should be increased. Currently, the article cites only 26 references, which is insufficient for a review of this scope. The inclusion of additional, recent, and directly relevant literature is strongly recommended.

Response: We agree with your comment. The manuscript has been revised to increase both its length and depth of coverage. The number of references has been expanded to 49, incorporating recent and directly relevant literature.

Many thanks

Reviewer 2 Report

Comments and Suggestions for Authors

This manuscript is an up-to-date and necessary review article, focusing on the environmental influence caused by diagnostic routes for colorectal cancer. In my opinion, this study has strong impact and may attract attention from gastroenterologists, public health staff, and medical managers. If the following points are addressed, the clarity, significance, and contribution of this research will improve further.

Major Comments

The manuscript says that new non-invasive examination tools (such as serum or stool biomarkers) need assessment about environmental influence. This part could be stronger if authors add some predictions or discussion about possible environmental problem points related to these new methods.

The article correctly points out that patient movement is a main cause of carbon output. However, deeper discussion is recommended, including the effect of concentrating specialized medical service, comparison between countryside and city medical access, and the usefulness of decentralized testing for reducing movement-related carbon emission.

Minor Comments

The explanation about UK and US cancer check programs is written both in Introduction section and Section 3. For smoother reading and to avoid repeating, it is better to combine this information into one part.

The reduction in conversion rate of the Urgent Suspected Cancer (USC) pathway is shown as "from 5.4 in 2011/12 to 2.8 in 2022/23." Please explain if these are percentage values, and make sure all units are properly described.

Several mistakes or inconsistencies can be found in the reference list, so authors should check references carefully.

Author Response

Thank you for your comments. Please find below responses

Major Comments

Comment: The manuscript says that new non-invasive examination tools (such as serum or stool biomarkers) need assessment about environmental influence. This part could be stronger if authors add some predictions or discussion about possible environmental problem points related to these new methods.

Response: We agree with your comment and have made substantial revisions to the discussion section to include predictions and talking points regarding potential environmental challenges associated with emerging non-invasive examination methods.

Comment: The article correctly points out that patient movement is a main cause of carbon output. However, deeper discussion is recommended, including the effect of concentrating specialized medical service, comparison between countryside and city medical access, and the usefulness of decentralized testing for reducing movement-related carbon emission.

Response: We agree with your comment and have revised the discussion section to incorporate a deeper analysis of travel-related emissions and urban-rural disparities in access.

Minor Comments

Comment: The explanation about UK and US cancer check programs is written both in Introduction section and Section 3. For smoother reading and to avoid repeating, it is better to combine this information into one part.

Response: We agree with your comment and have restructured the manuscript to consolidate the information on UK and US cancer screening programs, thereby improving flow and reducing repetition.

Comment: The reduction in conversion rate of the Urgent Suspected Cancer (USC) pathway is shown as "from 5.4 in 2011/12 to 2.8 in 2022/23." Please explain if these are percentage values, and make sure all units are properly described.

Response: We agree with your comment. The figures have been clarified as percentage values in the revised manuscript, and all units have been properly described.

Comment: Several mistakes or inconsistencies can be found in the reference list, so authors should check references carefully.

Response: We agree with your comment. The reference list has been thoroughly reviewed and corrected for any inconsistencies or errors.

Reviewer 3 Report

Comments and Suggestions for Authors

This article is an interesting and original narrative review about the environmental impact of the colorectal cancer screening methods. Diagnostic methods, such as FIT, colonography (CTC), and colonoscopy were discussed from an environmental point of view, showing significantly different environmental impacts. Despite the complexity of these qualitative and quantitative evaluations and the small number of retrievable papers coherently reporting data about this topic, this review is well done, well structured, and clearly explained, but lacks of some valuable details; therefore I have some issues to address to the authors in order to make this work more complete and more valuable:

  • I noticed that the primary source of environmental impact comes from the travel of medical staff and patients using internal combustion engine vehicles. I would like to know if the authors considered the case where many people are switching to electric or hybrid cars, as well as the public transports. This should considerably reduce the pollution due to internal combustion vehicles. No articles report the changes due to this phenomenon in the last years and/or a projection of this phenomenon in the next years? If confirmed, I recommend that the authors report this finding.
  • This paper lacks images. In my opinion an image/scheme about how these environmental assessments are done (for example to explain LCA or hybrid-LCA in a visual point of view) should be mandatory; this could improve the understanding of the evaluation methods here described.
  • No innovative techniques for colorectal cancer detection have been discussed here, such as the use of advanced and self-regenerating sensors and biosensors, which have shown promising results in detecting cancer by analyzing in vitro samples with minimal use of consumables and consuming few watts of energy. The following studies should be considered to support this addition:

https://doi.org/10.3390/bios15010056,  

https://doi.org/10.3390/nano13040674.

  • I am missing the actual conclusions of the work, since the conclusion section, at the current stage, seems to focus more on the study's limitations and difficulties about the not standardized methods to evaluate environmental impact, rather than providing real conclusions based on the data presented in the previous sections. I recommend that the authors include clearer and more quantitative conclusions.

Author Response

Many thanks for your comments. Please find below response to comments.

Comment- I noticed that the primary source of environmental impact comes from the travel of medical staff and patients using internal combustion engine vehicles. I would like to know if the authors considered the case where many people are switching to electric or hybrid cars, as well as the public transports. This should considerably reduce the pollution due to internal combustion vehicles. No articles report the changes due to this phenomenon in the last years and/or a projection of this phenomenon in the next years? If confirmed, I recommend that the authors report this finding.

Response -We agree with your comment. We have expanded the discussion section to address assumptions made in the reviewed literature regarding travel-related emissions. We now include considerations of the increasing use of electric and hybrid vehicles, as well as public transport, and discuss urban versus rural differences travel.

Comment- This paper lacks images. In my opinion an image/scheme about how these environmental assessments are done (for example to explain LCA or hybrid-LCA in a visual point of view) should be mandatory; this could improve the understanding of the evaluation methods here described.

Response- We agree with your comment. To improve clarity and understanding, we have included two images in the revised manuscript: one illustrating the stages of Life Cycle Assessment (LCA) and another depicting the Life Cycle Impact Assessment (LCIA) methodology.

Comment- No innovative techniques for colorectal cancer detection have been discussed here, such as the use of advanced and self-regenerating sensors and biosensors, which have shown promising results in detecting cancer by analyzing in vitro samples with minimal use of consumables and consuming few watts of energy. The following studies should be considered to support this addition:

https://doi.org/10.3390/bios15010056

https://doi.org/10.3390/nano13040674

Response- Although this review does not aim to provide an in-depth analysis of specific emerging diagnostic technologies, we agree with your comment. We have made substantial revisions to include a discussion on innovative techniques referencing one of the suggested studies and exploring how our findings may influence their clinical implementation.

Comment- I am missing the actual conclusions of the work, since the conclusion section, at the current stage, seems to focus more on the study's limitations and difficulties about the not standardized methods to evaluate environmental impact, rather than providing real conclusions based on the data presented in the previous sections. I recommend that the authors include clearer and more quantitative conclusions.

Response- We agree with your comment. The conclusion section has been thoroughly revised to ensure it presents clear and quantitative conclusions aligned with the aims and objectives of the review.

Round 2

Reviewer 1 Report

Comments and Suggestions for Authors

Accept

Reviewer 3 Report

Comments and Suggestions for Authors

This revised version of the manuscript has been significantly improved and enriched of essential contents and details. The authors have carefully addressed all the reviewer's comments, by adding images and clarifications (as requested), crucial to enhance the understanding of the article and of the conclusions, as well as of the reported data analysis techniques.

Given that, I believe that this manuscript is now suitable for publication in its current form.